# Prediction of coronary artery bypass graft outcomes using a single surgical note: An artificial intelligence-based prediction model study

**John Del Gaizo**[1], **Curry Sherard**[2], **Khaled Shorbaji**[1], **Brett Welch**[1], **Roshan Mathi**[1,2], **Arman Kilic**[1] *

**1** Division of Cardiothoracic Surgery, Department of Surgery, Medical University of South Carolina, Charleston, South Carolina, United States of America, **2** College of Medicine, Medical University of South Carolina, Charleston, South Carolina, United States of America

* kilica@musc.edu

## Abstract

### Background

Healthcare providers currently calculate risk of the composite outcome of morbidity or mortality associated with a coronary artery bypass grafting (CABG) surgery through manual input of variables into a logistic regression-based risk calculator. This study indicates that automated artificial intelligence (AI)-based techniques can instead calculate risk. Specifically, we present novel numerical embedding techniques that enable NLP (natural language processing) models to achieve higher performance than the risk calculator using a single preoperative surgical note.

### Methods

The most recent preoperative surgical consult notes of 1,738 patients who received an isolated CABG from July 1, 2014 to November 1, 2022 at a single institution were analyzed. The primary outcome was the Society of Thoracic Surgeons defined composite outcome of morbidity or mortality (MM). We tested three numerical-embedding techniques on the widely used TextCNN classification model: 1a) Basic embedding, treat numbers as word tokens; 1b) Basic embedding with a dataloader that Replaces out-of-context (ROOC) numbers with a tag, where context is defined as within a number of tokens of specified keywords; 2) Scale-Num, an embedding technique that scales in-context numbers via a learned sigmoid-linear-log function; and 3) AttnToNum, a ScaleNum-derivative that updates the ScaleNum embeddings via multi-headed attention applied to local context. Predictive performance was measured via area under the receiver operating characteristic curve (AUC) on holdout sets from 10 random-split experiments. For eXplainable-AI (X-AI), we calculate SHapley Additive exPlanation (SHAP) values at an ngram resolution (SHAP-N). While the analyses focus on TextCNN, we execute an analogous performance pipeline with a long short-term memory (LSTM) model to test if the numerical embedding advantage is robust to model architecture.

**Data Availability Statement:** Data cannot be shared publicly because of private health information. Data are available from the Medical

University of South Carolina Institutional Data Access / Ethics Committee for researchers who meet the criteria for access to confidential data. Specifically, even the deidentified dataset is only available through an appropriately executed data use agreement, upon approval from the Medical University of South Carolina Institutional Data Access / Ethics Committee. For data requests, the contact information for MUSC IRB is as follows: URL: https://research.musc.edu/resources/ori/irb-contacts Phone: 843-792-4148 Stacey Goretzka, CIP Program Manager 843-792-6527 goretzka@musc.edu Summer Young, JD, MPH, CIP IRB Reliance Manager 843-792-4144 youngsn@musc.edu A code example with mock data is provided in the associated code base at the following URL: https://github.com/musc-surgical-innovation-center/attntonum.

**Funding:** The author(s) received no specific funding for this work.

**Competing interests:** The authors have declared that no competing interests exist.

## Results

A total of 567 (32.6%) patients had MM following CABG. The embedding performances are as follows with the TextCNN architecture: 1a) Basic, mean AUC 0.788 [95% CI (confidence interval): 0.768–0.809]; 1b) ROOC, 0.801 [CI: 0.788–0.815]; 2) ScaleNum, 0.808 [CI: 0.785–0.821]; and 3) AttnToNum, 0.821 [CI: 0.806–0.834]. The LSTM architecture produced a similar trend. Permutation tests indicate that AttnToNum outperforms the other embedding techniques, though not statistically significant verse ScaleNum (p-value of .07). SHAP-N analyses indicate that the model learns to associate low blood urine nitrate (BUN) and creatinine values with survival. A correlation analysis of the attention-updated numerical embeddings indicates that AttnToNum learns to incorporate both number magnitude and local context to derive semantic similarities.

## Conclusion

This research presents both quantitative and clinical novel contributions. Quantitatively, we contribute two new embedding techniques: AttnToNum and ScaleNum. Both can embed strictly positive and bounded numerical values, and both surpass basic embeddings in predictive performance. The results suggest AttnToNum outperforms ScaleNum. With regards to clinical research, we show that AI methods can predict outcomes after CABG using a single preoperative note at a performance that matches or surpasses the current risk calculator. These findings reveal the potential role of NLP in automated registry reporting and quality improvement.

## Introduction

Artificial intelligence (AI) in healthcare has demonstrated utility in predictive analytics, imaging interpretation, data extraction, and reducing workload inefficiencies [1–4]; including AI applications specific to cardiac surgery [2, 3, 5–16]. Important drawbacks of AI include the need for large datasets, the potential for error especially in high-risk situations, the possibility of privacy violations and abuse, a lack of trust and understanding of AI among physicians and patients [1, 4, 5, 7, 17], and the need for use case transparency for patient consent.

The Society of Thoracic Surgeons (STS) risk models have served as a credible gold standard for cardiac surgical quality reporting for decades [4, 18, 19]. The STS risk calculator, which is a logistic regression model, achieves an AUC performance of approximately 0.76, with a 95% CI of (0.73, 0.79) [20]. Since some of the STS risk calculator inputs are manually extracted from unstructured data, these models require significant manual data extraction and entry. An automated AI would allow providers to spend more time on clinical duties.

Recent studies that utilize structured EHR data to predict coronary artery bypass grafting (CABG)-associated risk achieve similar [21] or superior [22, 23] performance to the STS risk calculator and identify high-risk predictors [21, 24]. We hypothesize that neural network models can complement these analyses by extracting information from unstructured data, such as patient notes. Whether more condensed data inputs such as a clinical note can serve to develop well performing risk models has yet to be explored [4, 10–16].

To answer this question, we evaluated several embedding techniques with convolutional neural network (CNN) and LSTM models for predicting post-CABG outcomes using a single preoperative surgical consult note, including a novel attention-based technique to embed

numerical tokens. Our results indicate that low parameter-count neural networks can achieve superior predictive performance to the current risk calculator standard, given utilization of the novel numerical embeddings and calibration of model output to a desired class-separation threshold.

## Materials and methods

### Data

Isolated CABG procedures performed from July 1, 2014 to November 1, 2022 at a single center were identified. These notes were all entered within 30 days of CABG. The notes consisted of inpatient consult notes with a "history of present illness" section. If multiple notes existed, the most recent to CABG was the only note included. According to a sample of our data, about 5.4% of the patient population is on-pump. This study was deemed exempt from review by the institutional review Board (IRB Pro00122587), the committee that oversees research ethics at the Medical University of South Carolina.

The notes occasionally listed symptoms in a bulleted format, but often did not. Important metrics recorded in the notes included creatinine and blood urine nitrate (BUN) values, commonly in the format of "BUN 24.0 (H)" or "creatinine .7 (L)". The healthcare professional transcribed low (L) for the BUN value if it was less than or equal to 20 and high (H) otherwise. The analogous cutoff for creatinine was 1.0.

### Primary outcome

The primary outcome was the STS-defined composite outcome of operative mortality or major morbidity (MM). Operative mortality is defined by the STS as mortality occurring within 30 days of surgery or as an inpatient during the index hospitalization. Major morbidities included postoperative stroke, acute renal failure, cardiac reoperation, deep sternal wound infection, and prolonged ventilation.

### Predictive performance experiments

We ran 10 experiments of our machine learning pipeline. Each experiment left a random 15% (261 samples) of the data for a holdout set, and then split the remaining notes 85%/15% (1,254/222) for training and validation. The splits were stratified to ensure a similar ratio of MM subjects between the training, validation, and holdout sets. For each experiment, the model was fit on the 1,254 training samples. The same 10 split sets were employed to compare different embedding techniques and models to ensure consistent comparisons.

For each experiment, we calibrate the validation set probability of MM predictions to the observed MM rate via a linear regression to obtain a calibration slope and intercept. Specifically, the validation set subjects are grouped according to MM prediction, binned at 0.1 resolution, and the mean MM rate serves as the dependent variable. The fit linear regression model is then applied to calibrate holdout set predictions. The performance was measured by AUC on the holdout sets for each of the 10 experiments. Standard textual preprocessing was performed, such as regular expressions to remove dates.

### Convolutional neural networks

Our base CNN architecture is TextCNN [25]. Previous research indicates that TextCNN shows comparable classification performance on small medical datasets to much larger models such as BERT [10, 26], but with orders of magnitude less memory, thereby enabling faster prototyping and experimentation.

The hyperparameters are as follows: learning rate of $3.5^{-4}$, batch size of 64, 44 kernels per convolutional filter layer, embed dimension of 50, 4 filter layers of sizes (1, 2, 3, 5), 100 epochs, 0.5 dropout applied to the final fully connected layer, and a length cut off of 3,000 tokens with 0 padding for smaller notes. This results in a total of 176 = 44x4 convolutional filters.

## Long Short-Term Memory (LSTM) neural networks

The base LSTM architecture consists of the same architecture as the presented TextCNN, except the convolutional layer is replaced with a bi-directional LSTM (Bi-LSTM) layer. We refer to this model as TextLSTM. Similar to TextCNN, a max pool is applied temporally before the classification layer. However, the max pool is applied across the bi-LSTM's cell output states instead of convolutional filter activations.

We configure the TextLSTM architecture to have the same layer dimensions as TextCNN, except the convolutional layer is replaced with a Bi-LSTM. The LSTM hidden dimension is 88, which means the output dimension of the Bi-LSTM is 176 by 3000 (sequence length), and this is reduced to 176 after the max pool; dropout is 0.0; and the number of epochs is 50 instead of 100 as we found the LSTM model overfits more readily and that dropout does not provide regularization. Note that the hidden dimension size of 88 was chosen so that the classification layer would be of the same dimension as TextCNN's classification layer, 176.

## Embedding numerical information

Encoding numerical information is a challenging task in natural language processing [27]. Even advanced models treat numbers as text and encode them in a similar manner. This means that closely related numbers (i.e. 1.7 and 1.8) appear as 2 separate tokens, where similarity in the model output results from similarity in the semantic space. This poses a challenge for models that are trained on small datasets that may have few or no training samples with a hold-out sample's numerical value.

As a first step, we experiment with replacing uninformative numbers with a tag. Basic exploratory analyses on the dataset indicated that blood urine nitrate (BUN) and creatinine values listed in the notes strongly correlated with MM. Due to the nature of how the physicians transcribe notes, BUN/creatinine values are listed within close range to the key-terms "BUN" or "creatinine". A common format is "[BUN|creatinine] [numerical value] ([H|L])", where (H) means the physician labeled the value as high and (L) as low. We define context as tokens within 2 tokens to the right, or 1 token to the left of either BUN or creatinine. These are hyperparameters that can be adjusted.

Out-of-context numbers (OOC) are numerical tokens outside 2 tokens to the right or 1 token to the left. OOC numbers are replaced with the token "_INUM_" or "_lgnum_", if they are less than or greater than 1000, respectively (Fig 1). This substitution removes sporadic numerical tokens, thereby reducing the embedding matrix dimensionality while retaining BUN and creatinine signal.

We ran experiments with three embedding techniques: 1a) Basic embedding, treat numbers as word tokens; 1b) Basic embedding with a dataloader that Replaces Out-Of-Context numbers with a tag (ROOC); 2) ScaleNum, an embedding technique that scales in-context numbers via a sigmoid of a learned linear-log function; 3) AttnToNum, a ScaleNum-derivative that updates the ScaleNum embeddings via multi-headed attention applied to local context (Fig 1).

## ScaleNum

ScaleNum is an embedding technique we developed for numbers that are within context of the tokens "BUN" or "creatinine". ScaleNum scales the number to a multi-dimensional vector via

**Fig 1. Embedding pipeline for the example text "Cl 93.0 (L) 03/04/18 creatinine 5.2 (H) 03/05/18", for the TextCNN architecture model with AttnToNum embeddings, 10th run.** The numerical values correspond to the first dimension of the dimension-50 embeddings.

function, $g(x)$. $g(x)$ first clamps the number between 1 and 1,000, followed by the log function, then a linear layer from dimension 1 to embedding dimension (50), and a sigmoid:

$$y_j = g_j(x) = \sigma\left(a_j \times \log(\text{clamp}(x, [1, \ 1000])) + b_j\right)$$

$$\boldsymbol{y} = g(x) = \sigma(\boldsymbol{a} \times \log(clamp(x, [1, \ 1000]) + \boldsymbol{b}))$$

Where **y**, **a**, and **b** are vectors of size embedding dimension.

As BUN and creatinine values are frequently greater than 1 and always less than 1000, the clamp removes little numerical information while ensuring stability if there is a number outside this range. Note that g(x) is equivalent to $x^a e^b/(x^a e^b + 1)$ since $e^{a\log(x)+b} = x^a e^b$; where $x^a e^b$ is element-wise multiplied. The model learns multiple g(x) transformations, and then adds the resultant embeddings together. For this research, the model learns 5 g(x) transformations.

$$\widehat{g(x)} = \sum_1^5 g^k(x)$$

### AttnToNum

*AttnToNum* employs multi-headed self-attention between *ScaleNum's* $\widehat{g(x)}$ embeddings and the token embeddings within context to generate context-aware number embeddings (Fig 1). This research uses 5 attention heads, with a head dimension of 50/5 = 10.

$$e(x) = multi\_attn\left(\widehat{g(x)}\right)$$

### Embedding-comparison permutation tests

Permutation tests were used to verify if a given numeric-embedding technique statistically significantly improves performance. The 4 techniques are Basic, Basic with Replace Out-Of-Context numbers (ROOC), ScaleNum, and AttnToNum, for a total of 6 comparisons: (1) AttnToNum vs ScaleNum, (2) AttnToNum vs ROOC, (3) ROOC vs Basic, (4) ScaleNum vs ROOC, (5) ScaleNum vs Basic, and (6) ROOC vs Basic. For each comparison, a length-10 AUC gap vector, *gv*, is calculated as the difference between a given model's holdout AUC

values, $A$, and a reference model's holdout AUC values, $T$.

$$gv = A - T$$

The sum of the elements of $gv$ is stored as the true gap value:

$$t = sum(gv)$$

For each of 35,000 iterations, we generate a length-10 vector composed of random samples from 2 values [1, –1], $sn$, and take the dot product of $sn$ with $gv$ to obtain a mock gap value, $m$. This is equivalent to randomly flipping the sign of each element of $gv$, and taking the sum:

$$m = sn \cdot gv$$

The true gap, $t$, is compared against the 35,000 mock gaps, $m$, to obtain 1-sided and 2-side p-values:

$$p_1 = (\text{sum}(t \leq m) + 1)/1e4$$

$$p_2 = (\text{sum}(t \leq abs(m)) + 1)/1e4$$

## Embedding correlations

To test the hypothesis that the model learns to incorporate both number magnitude and local context to derive a numerical semantic space, we calculate the correlations between pre-convolution embeddings associated with the numerical embeddings for a series of BUN and creatinine values. For example: the correlation between the numerical embeddings for "BUN 10.0 (L)" and "creatinine 4.0 (H)". The pipeline to generate the embedding correlations is as follows:

1. Generate 2 phrases, each in the format of "[BUN|creatinine] [value] ([L|H])". If the BUN value is higher than 20, it is labeled as High (H) and low (L) otherwise. For creatinine, this cutoff was 1.0

2. Pass each phrase to the TextCNN model's embedding layers to obtain the pre-attention and post-attention embeddings for each phrase. The specific TextCNN instance is selected based on the experiment with the best performance.

3. Extract the embedding vector that corresponds to the number token ([value]) for each phrase, and then calculate the correlation coefficient between the two vectors.

## ngram importance with SHAP-N

We employ an ngram-resolution version of SHapley Additive ExPlanations (SHAP) analysis to identify individual ngram contributions to sample predictions. We term this analysis as SHAP-Ngram resolution (SHAP-N). These contributions can be directly calculated from the TextCNN architecture.

Via the SHAP approach, the aim is to find the simplified, sample-specific, additive model that equals the original model for a given sample, $s$.

$$logit_s = g_s = \phi_0 + \sum_F \phi_{f,s} \tag{1}$$

Each $\phi_{f,s}$ equals the relative logit contribution associated with feature $f$ for sample $s$, where relative contribution is defined as the logit contribution for $s$ from feature $f$, minus the mean logit contribution for feature $f$ in the dataset:

$$\phi_{f,s} = logit_{s,f} - \mu_f \tag{2}$$

Finally, $\phi_0$ equals the sum of the mean logit contributions. Note that $\phi_0$ is sample-independent:

$$\phi_0 = \sum_F \mu_f \tag{3}$$

In the TextCNN architecture, each convolutional filter contributes only one activation per sample due to the sample-wide max pool after the convolutional layer, and this activation is multiplied with a single coefficient in the fully connected layer. These scaled activations, one per convolutional filter, are additively combined to create a logit [25]. Therefore, the logit for a sample can be calculated as:

$$logit_s = w_0 + \sum_F w_f a_{s,f} \tag{4}$$

Where "$F$" is the number of filters, and each filter has one and only one logit contribution per sample.

Eq 4 can be reformulated as Eq 1 by subtracting the mean value for each filter:

$$\begin{aligned} logit_s &= w_0 + \sum_F w_f \mu_f + \sum_F w_f \left( a_{s,f} - \mu_f \right) \\ &= \phi_0 + \sum_F \phi_{f,s} \end{aligned} \tag{5}$$

Where $\phi_0 = w_0 + \sum_F w_f \mu_f$, and $\phi_{f,s} = w_f (a_{s,f} - \mu_f)$ Therefore, the SHAP value for each filter, $f$, and sample, $s$, combination equals $w_f (a_{s,f} - \mu_f)$.

The $\phi_{f,s}$ values directly indicate the importance of each passing ngram **_if_** each filter has a distinct ngram. However, 2 common scenarios violate this assumption: (1) the same ngram passes multiple max pools, and (2) sub-ngrams of a passing ngram will pass other max pools. For both scenarios, we add the SHAP values together: $\phi_{n,s} = \sum_{F \in n} \phi_{f,s}$, where $n$ represents the ngram, and $F \in n$ represents the ngrams that equal $n$, or are sub-ngrams of $n$. If $n$ only passes one max pool and has no sub-ngrams passing other filters, then $\phi_{n,s} = \phi_{f,s}$. Therefore, Eq 5 can be reformulated as follows, where we define the importance value for ngram, $n$, and subject, $s$, as $\phi_{n,s}$, and $N$ is the number of ngrams:

$$logit_s = \phi_0 + \sum_N \phi_{n,s} \tag{6}$$

Finally, Zhao et al [28] set-union the overlapping ngrams and sum together the associated $\phi_{n,s}$ values to create a new set of $N'$ features:

$$logit_s = \phi_0 + \sum_{N'} \phi_{n',s} \tag{7}$$

However, we found that this leads to a smoothing effect of long overlaps that are hard to interpret on our dataset. Therefore, we leave the resolution at the ngram level (Eq 6), and refer to the importances, $\phi_{n,s}$ as SHAP-N (SHAP-Ngram) values.

We calculate the final SHAP-N for an ngram as its average SHAP-N value across all the samples in the dataset (holdout or validation):

$$\phi_n = 1 \Big/ |S| \sum_s \phi_{n,s} \tag{8}$$

We present the top SHAP-N values for the highest performant experiment, as well as the SHAP-N values for ngrams that contain or consist of a BUN/creatinine token followed by a number.

## Results

A total of 1,738 patients who underwent isolated CABG during the study period had a preoperative surgical consult note within 30 days of surgery with a "history of present illness" section and therefore met inclusion criteria. Of the 1,738 subjects, 567 (32.6%) had the STS-defined outcome of operative mortality or major morbidity. Of the 567 MM patients, 515 had a major morbidity only, 13 had only an operative mortality without major morbidity, and 39 experienced both major morbidity and mortality.

### Predictive performance experiments

For the TextCNN architecture, the classification performances are as follows (Fig 2):

1a) Basic: mean AUC of $0.788 \pm 1.23 \times 10^{-3}$ (95% confidence interval (CI): 0.768–0.809)

1b) ROOC: $0.801 \pm 5.74 \times 10^{-4}$ (CI: 0.788–0.815)

2) ScaleNum: $.808 \pm 8.34 \times 10^{-4}$ (CI: 0.785–0.821)

3) AttnToNum: $0.821 \pm 6.30 \times 10^{-4}$ (CI: 0.806–0.834).

The holdout AUCs for the 10 experiments with TextCNN are shown in Table 1.
The TextLSTM architecture performances are as follows:

1a) Basic, mean AUC of $0.771 \pm 3.27 \times 10^{-4}$ (CI: 0.770–0.791)

1b) ROOC, $0.798 \pm 5.59 \times 10^{-4}$ (CI: 0.783–0.811)

2) ScaleNum, $0.817 \pm 6.10 \times 10^{-4}$ (CI: 0.798–0.829)

3) AttnToNum, $0.825 \pm 8.06 \times 10^{-4}$ (CI: 0.809–0.843).

The holdout AUCs for the 10 experiments with TextLSTM are shown in Table 2.
The parameter counts for the TextCNN models are shown in Table 3. TextLSTM has the same parameter counts for most layers, except "Post-Embed Layers" has 98,737 parameters. Which results in a total parameter count increase of 74,184 (98,737–24,553).

For TextCNN with AttnToNum embeddings, the mean calibration slope across the 10 experiment runs is 0.930±0.079, and the mean calibration intercept is 0.056±0.035.

For TextLSTM with AttnToNum embeddings, the mean calibration slope across the 10 experiment runs is 1.007±0.090, and the mean calibration intercept is 0.011±0.042.

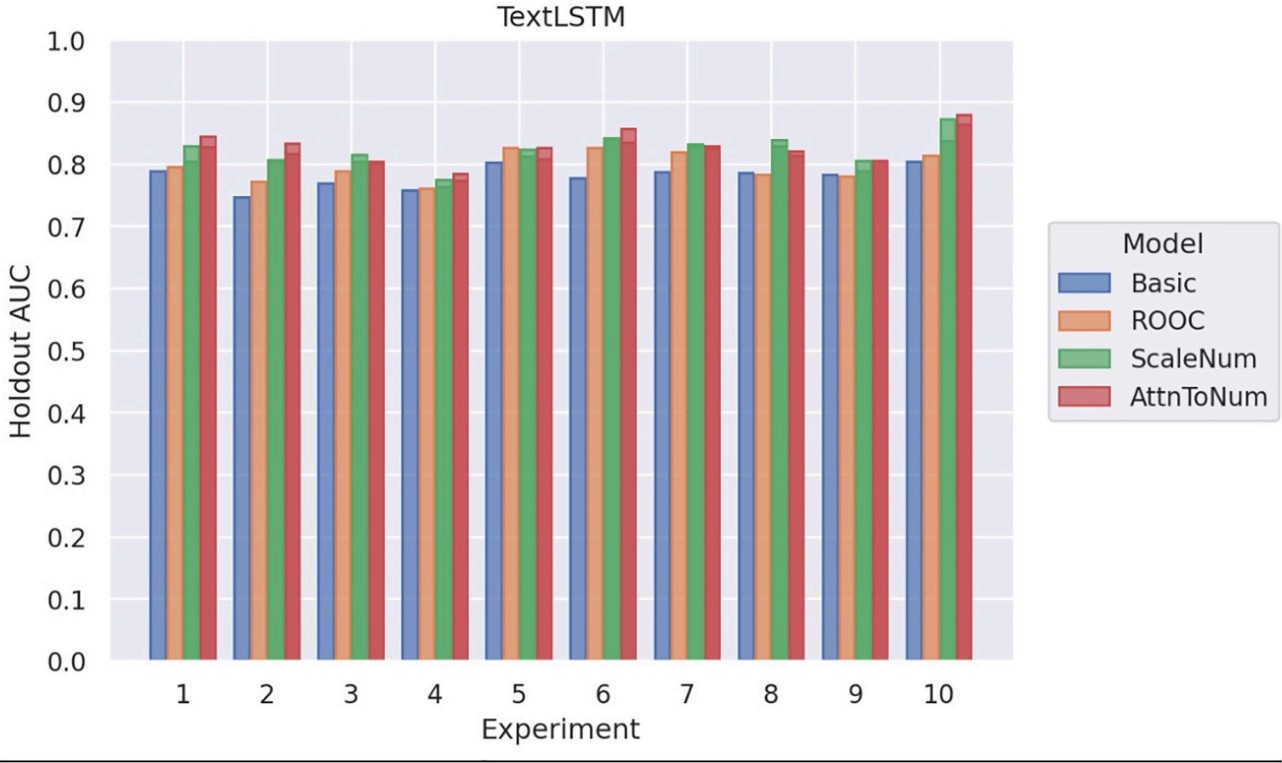

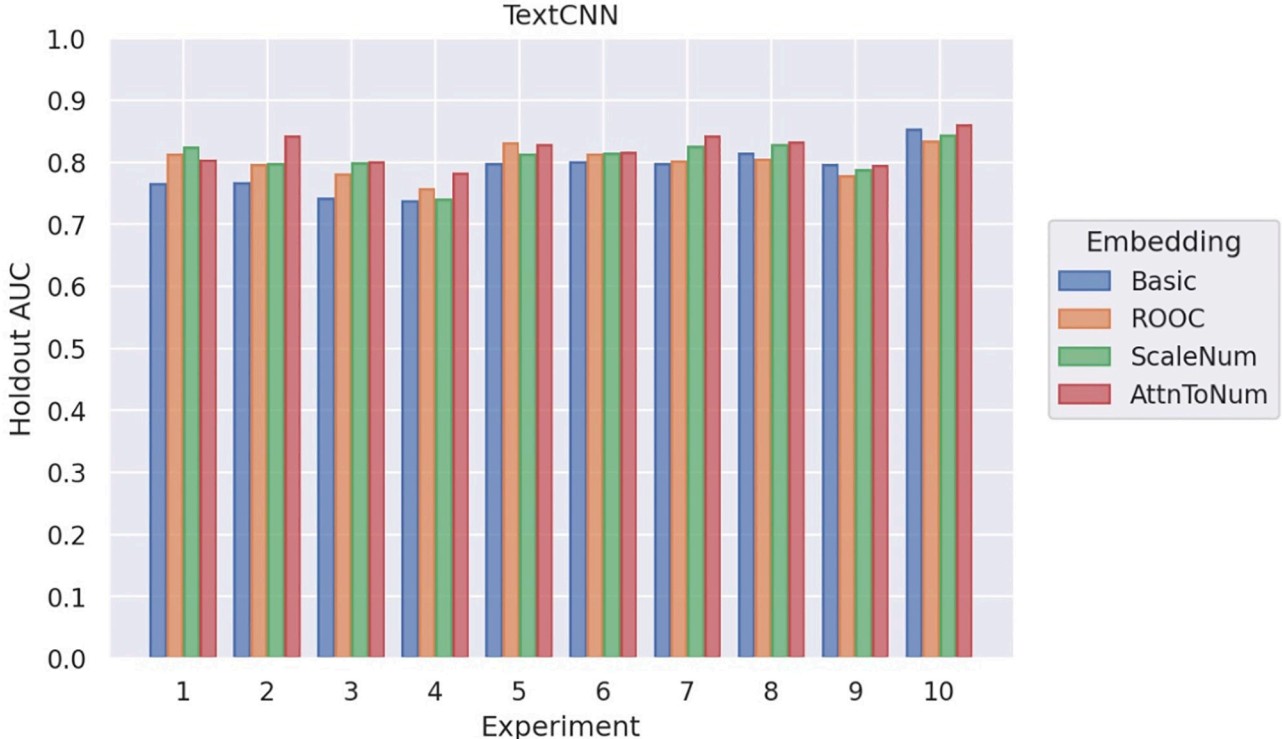

**Fig 2. Holdout AUCs for the 10 experiment splits.**

**Table 1. TextCNN Holdout AUCs.** Holdout AUCs for TextCNN Experiments.

| Experiment | 1 | 2 | 3 | 4 | 5 | 6 | 7 | 8 | 9 | 10 |
|---|---|---|---|---|---|---|---|---|---|---|
| Basic | 0.766 | 0.767 | 0.742 | 0.738 | 0.798 | 0.800 | 0.799 | 0.815 | 0.797 | 0.854 |
| ROOC | 0.813 | 0.796 | 0.782 | 0.758 | 0.832 | 0.813 | 0.802 | 0.805 | 0.778 | 0.834 |
| ScaleNum | 0.825 | 0.798 | 0.799 | 0.741 | 0.813 | 0.815 | 0.826 | 0.828 | 0.789 | 0.845 |
| AttnToNum | 0.804 | 0.843 | 0.800 | 0.782 | 0.829 | 0.816 | 0.843 | 0.833 | 0.795 | 0.860 |

**Table 2. TextLSTM holdout AUCs.** Holdout AUCs for TextLSTM experiments.

| Experiment | 1 | 2 | 3 | 4 | 5 | 6 | 7 | 8 | 9 | 10 |
|---|---|---|---|---|---|---|---|---|---|---|
| Basic | 0.789 | 0.748 | 0.770 | 0.760 | 0.804 | 0.778 | 0.789 | 0.787 | 0.785 | 0.805 |
| ROOC | 0.797 | 0.773 | 0.789 | 0.761 | 0.827 | 0.827 | 0.821 | 0.784 | 0.781 | 0.815 |
| ScaleNum | 0.830 | 0.808 | 0.816 | 0.765 | 0.813 | 0.842 | 0.830 | 0.840 | 0.790 | 0.838 |
| AttnToNum | 0.846 | 0.834 | 0.802 | 0.774 | 0.827 | 0.836 | 0.826 | 0.816 | 0.807 | 0.881 |

**Table 3. Parameter counts.** The parameter counts for the TextCNN layers. TextLSTM has the same parameter counts, except "Post-Embed Layers" has 98,737 parameters instead of 24,553 for a difference of 74,184 parameters.

| Number Embed Type | Embed | Number Embedding | Attention | Post-Embed Layers (CNN) | Total |
|---|---|---|---|---|---|
| Basic | 550,550 (11,011x50) | 0 | 0 | 24,553 | 575,103 |
| ROOC | 475,900 (9,518x50) | 0 | 0 | 24,533 | 500,453 |
| ScaleNum | 470,400 (9,408x50) | 600 | 0 | 24,553 | 495,553 |
| AttnToNum | 470,400 (9,408x50) | 600 | 10,200 | 24,553 | 505,753 |

## Embedding-comparison permutation tests

For TextCNN and TextLSTM, the permutation tests produced nearly identical 2-sided p-values as shown in Table 4.

## Embedding correlations

Pre-attention (ScaleNum) and post-attention (AttnToNum) embedding correlations are shown in Fig 3. The embeddings are obtained from the AttnToNum embedding layer in experiment 10's TextCNN.

As an illustrative example, Table 5 utilizes the same TextCNN instance to obtain correlations between number embeddings for similar phrases in each stage of the AttnToNum embedding.

**Table 4. Permutation tests.** Permutation tests with TextCNN and TextLSTM.

| | TextCNN | TextLSTM |
|---|---|---|
| Experiment | p | p |
| AttnToNum vs ScaleNum | 0.067 | 0.066 |
| AttnToNum vs ROOC | 0.012 | 0.012 |
| AttnToNum vs Basic | 0.0038 | 0.0036 |
| ScaleNum vs ROOC | 0.18 | 0.18 |
| ScaleNum vs Basic | 0.022 | 0.022 |
| ROOC vs Basic | 0.106 | 0.112 |

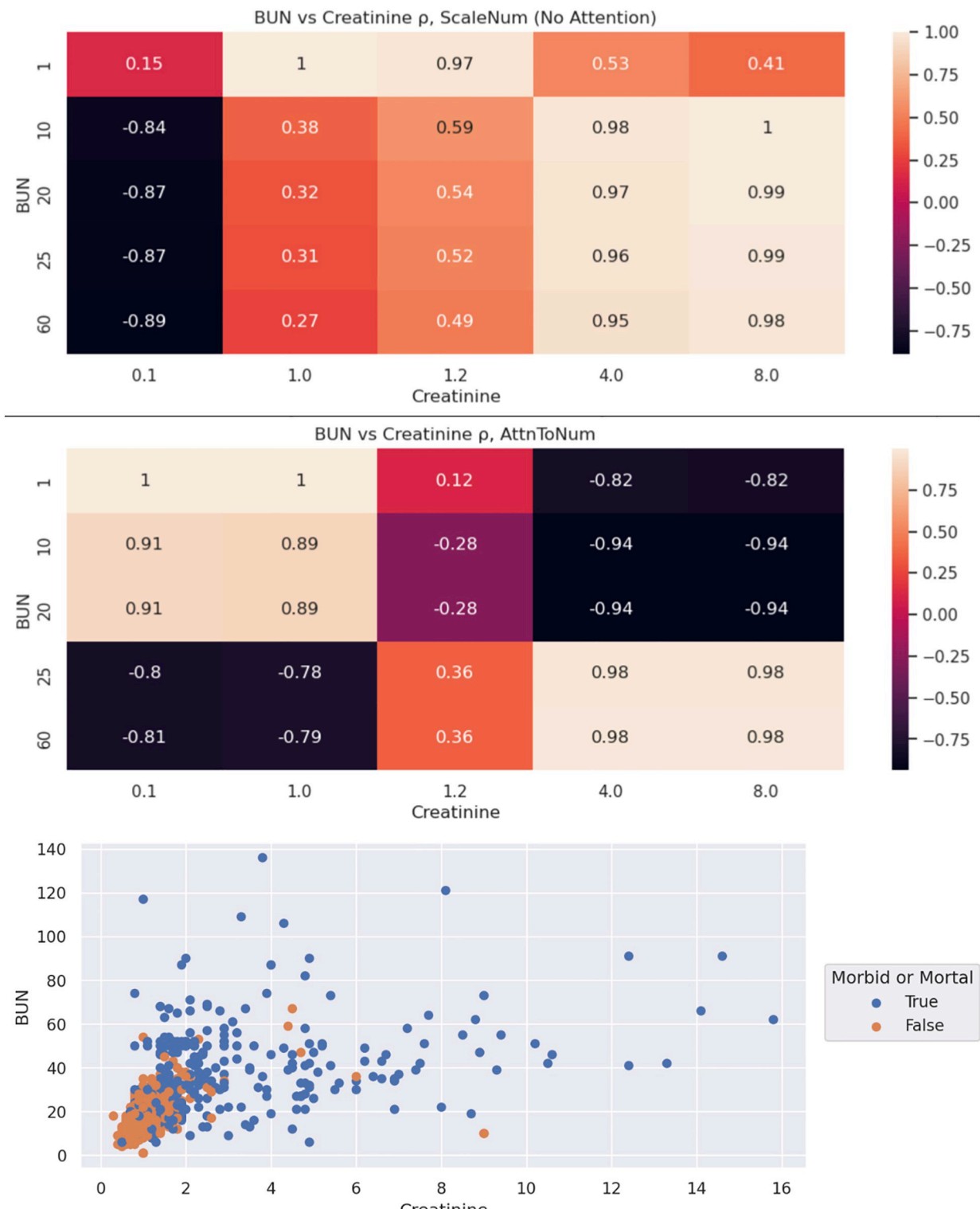

**Fig 3. BUN & creatinine embedding correlations.** The bottom subplot of BUN vs creatinine values extracted from the text reveal why some embeddings are nearly identical, such as "creatinine 4.0 (H)" & "creatinine 8.0 (H)".

**Table 5. Embedding stage and correlation.** Embedding correlations between extracted numbers in each step of the AttnToNum pipeline: i.e. correlations between the embeddings for 3.4 and 3.5, and between 25.0 and 3.8. Note that 3.4 is a low BUN value, 3.8 is a high creatinine value, and 25.0 is a high BUN value.

| Phrases | ROOCρ | ScaleNum ρ | AttnToNum ρ |
|---|---|---|---|
| "BUN 3.4 (L)", "Creatinine 3.8 (H)" | 1.0 | 0.999 | -0.876 |
| "BUN 3.4 (L)", "Creatinine 3.8 (L)" | 1.0 | 0.999 | -0.638 |
| "BUN 25.0 (H)", "Creatinine 3.8 (H)" | 1.0 | 0.961 | 0.983 |
| "BUN 25.0 (H)", "Creatinine 3.8 (L)" | 1.0 | 0.961 | 0.671 |

**Table 6. Features of importance.** Top 10 features of importance for experiment 10, the best performing experiment for AttnToNum with TextCNN. The "_date_" tag replaced dates.

| SHAP | ngram |
|---|---|
| 0.241 | Creatinine 4.7 (H) _date_ glucose |
| 0.208 | 4.7 (H) _date_ Creatinine 4.7 |
| 0.189 | Creatinine 4.7 (H) _date_ BUN |
| 0.188 | Creatinine 4.5 (H) _date_ BUN |
| 0.184 | Creatinine 2.5 (H) _date_ glucose |
| 0.181 | Creatinine 3.6 (H) _date_-bun |
| 0.179 | Creatinine 7.4 (H) _date_-bun |
| 0.151 | Creatinine 2.3 (H) _date_-bun |
| 0.150 | _date_ Creatinine 4.7 |
| 0.149 | _date_ Creatinine 4.5 |

## ngram importance with SHAP-N

The top 10 features of importance for the best performing experiment for AttnToNum with TextCNN, experiment 10, are shown in Table 6.

Fig 4 shows a plot of SHAP-N values for different numerical values of BUN and creatinine. The SHAP-N values are associated with trigrams that follow the pattern of "[BUN|creatinine]

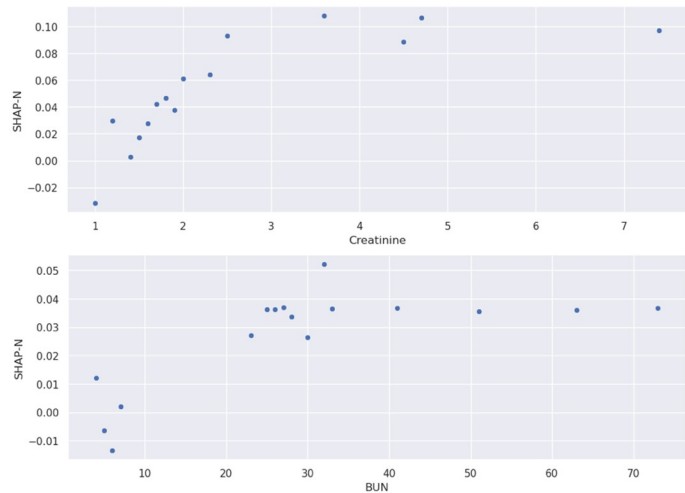

**Fig 4.** Top to bottom: Experiment 10, SHAP-N values associated with (a) creatinine and (b) BUN values extracted from ngrams ("BUN 1.1 (L)"). Note: the embeddings are context-based, the values will not always be the same for each ngram. The presented SHAP values correspond to trigrams that follow the format of "[BUN|creatinine] [value] ([L| H])".

**Table 7. Attention weights.** The attention weights for TextCNN (AttnToNum embeddings) of the 10th experiment for phrases of the template: "_date_ *(BUN|creatinine)* 6.3 *(H|L)*." The 2nd head focuses on the number's magnitude if the physician labeled the test as high, (H). The 4th and 5th heads focus on physician label and _date_, respectively. The 3rd head focuses on test label if the label is low, (L). The 1st head focuses on metric if the metric is creatinine.

| Head | _date_ | creatinine | 6.3 | (H) |
|---|---|---|---|---|
| 1 | 0.01 | **0.99** | 0.0 | 0.0 |
| 2 | 0.0 | 0.0 | **1.0** | 0.0 |
| 3 | **1.0** | 0.0 | 0.0 | 0.0 |
| 4 | 0.0 | 0.0 | 0.0 | **1.0** |
| 5 | **1.0** | 0.0 | 0.0 | 0.0 |

| Head | _date_ | creatinine | 6.3 | (L) |
|---|---|---|---|---|
| 1 | 0.01 | **0.98** | 0.0 | 0.01 |
| 2 | 0.0 | 0.0 | 0.03 | **0.97** |
| 3 | 0.0 | 0.0 | 0.0 | **1.0** |
| 4 | 0.0 | 0.0 | 0.0 | **1.0** |
| 5 | **0.99** | 0.0 | 0.0 | 0.01 |

| Head | _date_ | BUN | 6.3 | (H) |
|---|---|---|---|---|
| 1 | **0.99** | 0.0 | 0.0 | 0.0 |
| 2 | 0.0 | 0.0 | **1.0** | 0.0 |
| 3 | **1.0** | 0.0 | 0.0 | 0.0 |
| 4 | 0.0 | 0.0 | 0.0 | **1.0** |
| 5 | **1.0** | 0.0 | 0.0 | 0.0 |

| Head | _date_ | BUN | 6.3 | (L) |
|---|---|---|---|---|
| 1 | **0.43** | 0.0 | 0.00 | **0.57** |
| 2 | 0.0 | 0.03 | 0.03 | **0.97** |
| 3 | 0.0 | 0.0 | 0.0 | **1.0** |
| 4 | 0.0 | 0.0 | 0.0 | **1.0** |
| 5 | **0.99** | 0.0 | 0.0 | 0.01 |

[value] ([L|H])". The specific model instance is the TextCNN model trained during experiment 10.

## Attention visualizations

We present attention weights for each of the 5 attention heads for the best-performing model, TextCNN with AttnToNum embeddings from experiment 10, for the phrases: "_date_ creatinine 6.3 (H)", "_date_ creatinine 6.3 (L)", "_date_ BUN 6.3 (H)", and "_date_ BUN 6.3 (L)" (Table 7).

## Discussion

This study demonstrates the ability of artificial intelligence (AI) to predict the MM outcome after CABG using a small sample of the unstructured data in the EHR, a preoperative surgical consult note. The presented techniques surpass the baseline Society of Thoracic Surgeons (STS) model, which is among the most credible traditional risk assessment models [4, 18, 19]. According to a recent meta-analysis, the STS risk calculator achieves an AUC performance of 0.76, with a 95% CI of (0.73, 0.79) [20]. In addition, some of the STS risk calculator inputs are

manually extracted from unstructured data. An automated AI enables providers to spend more time on clinical duties.

Several recent analyses identify CABG-associated risk from structured EHR data [21–24]. Two of the analyses achieve similar or superior performance to our analysis with AUCs of 0.811 [22] and 0.831 [23] for post-CABG mortality and bleeding. These previous results and this own study's performance indicates that there is relevant signal in both structured and unstructured EHR data, we hypothesize that a model that combines both will surpass models trained on either modality alone.

Our pipeline embeds numerical magnitude. In comparison, standard tokenization treats similar magnitude numbers as two unique tokens. If the sample size is large, this may not be an issue since the model can learn the semantic space through training set encounters with similar context or exact exposures to the numbers. Small sample sizes do not provide as thorough a semantic space, and therefore the model may not learn to treat similar magnitude numbers in a similar manner nor encounter as many numbers in training. This numerical scaling capability to treat similar magnitude numbers in a similar manner is provided by ScaleNum.

Furthermore, context is important: "BUN 12.0" and "creatinine 12.0" carry nearly opposite meanings. We therefore developed AttnToNum, a localized attention technique that updates the numerical embeddings by incorporating local context. The correlations between post-attention embeddings confirm that context is learned, as shown in Table 5 (Embedding Stage and Correlations) and Fig 3. Both Table 5 and Fig 3 indicate that similar magnitude numbers can lead to post-attention embedding vectors with a negative correlation, even if the pre-attention correlation is close to 1 due to the similar magnitude. As a specific example, Table 5 shows that while ScaleNum outputs nearly identical numerical embeddings for 3.4 and 3.8 in the phrases "BUN 3.4 (L)" and "creatinine 3.8 (H)", AttnToNum outputs embeddings that capture the opposite meanings of these numbers (3.4 is a low BUN value, but a high creatinine value) with embedding vectors in nearly opposite direction (correlation of -.876). Table 7 (Attention Weights) indicates that the multi-headed attention applies different heads to learn context and magnitude. The 2nd attention head learns to focus on number magnitude if the label is (H). The other heads focus on context. If the label is low (L), then the attention heads do not focus on magnitude, suggesting that low BUN and creatinine values will have the same or similar embedding vectors with other low but different BUN and creatinine values. As a further confirmation, Fig 4 shows that the model (1) associated high BUN/creatinine values with MM; and (2) can interpret which number magnitudes are *relatively* high dependent on context.

In terms of performance, both AttnToNum and ScaleNum statistically significantly outperform basic tokenization. ScaleNum does not statistically significantly outperform Replacement of Out-of-Context tokens (ROOC), and AttnToNum does not statistically significantly outperform ScaleNum. However, the mean AUC, confidence intervals, and simulation 2-sided p-values still indicate a performance difference (Fig 2). Therefore, this embedding technique may benefit the medical research community as a baseline model to replace standard tokenization under certain scenarios.

Despite the extra layers, AttnToNum and ScaleNum have ~15% less parameters than Basic, see Table 3 (Parameter Counts). This results from the decrease in unique terms; numbers are not tokenized for AttnToNum and ScaleNum. The term reduction leads to less rows of the parametric-heavy embedding layer matrix. Most of this reduction results from the removal of out-of-context numbers, as shown by the much lower parameter count for ROOC than Basic.

A slight increase in model complexity (~600 parameters, 0.1% increase) with the inclusion of number embedding enables the model to capture numeric information by learning parameters for a scaling function that outputs similar embeddings for similar magnitude numbers.

The learned scaling function, $g(x) = \sigma(a\,log(x) + b) = x^a e^b/(x^a e^b + 1)$, is similar to a sigmoid in that it compresses numbers to a value between 0 and 1. However, in this case the sigmoid is applied to a linear-log. The learned parameters adjust the rate of change, $a$, and when to start increasing, $b$. Therefore, different parameters capture meaning from different scale numbers; this multi-scale information is available for downstream layers through the summation of the 5 $g(x)$ vectors into a single vector.

A more significant increase in model complexity via an attention layer (~10,000 parameters, 2%) also led to a performance increase. We hypothesize that the attention mechanism with local context informs the model which scale of the numerical embedding is relevant.

Despite an ability for LSTM models to embed long-range context, TextCNN and TextLSTM achieved similar performance. We hypothesize that this results from a property of the notes that related concepts are located close together, nullifying the long-term memory capabilities of LSTM. For example, symptoms, findings, etc. are often listed in a bulleted format.

Finally, the SHAP-N values for numerical BUN and creatinine ngrams also cohered with expertise knowledge, both high BUN and creatinine values are associated with MM.

In summary, this study represents promising evidence that AI-based automated approaches can be capable of predicting CABG risk and may eventually replace traditional risk models. We contribute 2 new embedding techniques for numerical values, AttnToNum and ScaleNum, and straightforward steps to calculate TextCNN SHAP values for ngrams.

## Limitations

We did not test transformer models, a class of models which includes BERT and other large language models (LLM). Such comparative experiments would likely reveal key insights, but the scope of this paper is already wide and the different techniques and prompts to train these advanced models can form a large body of analyses. Therefore, we aim to analyze possible performance benefits of including our numerical embeddings with LLMs in future research.

Our numerical embedding operates only on strictly positive numbers due to the log function. Another implementation could overcome this issue–for example, by clamping the minimum value and adding an offset, or by using a scaling function that is not log.

The hardcoded context hyperparameters that specify the number of left and right tokens correspond to an apriori injection of domain knowledge into the model. Ideally, these would not be hyperparameters, but attention weights learned from the data. For small clinical datasets, this apriori guidance can boost performance. For large datasets with higher-dimension embeddings, it may be possible to train more robust attention-transformations.

## Reproducibility

Due to the sensitive PHI nature of the data, we release the code to train models on a simulated dataset sampled from corpora provided by the Python NLTK library [29]. The code can be found here.

## Acknowledgments

This research was performed as part of the employment of the authors at the Medical University of South Carolina.

## Author Contributions

**Conceptualization:** John Del Gaizo, Khaled Shorbaji, Roshan Mathi, Arman Kilic.

**Data curation:** John Del Gaizo, Khaled Shorbaji, Roshan Mathi.

**Formal analysis:** John Del Gaizo, Curry Sherard, Khaled Shorbaji, Roshan Mathi.

**Investigation:** John Del Gaizo, Khaled Shorbaji, Roshan Mathi, Arman Kilic.

**Methodology:** John Del Gaizo, Khaled Shorbaji, Roshan Mathi, Arman Kilic.

**Project administration:** Curry Sherard, Khaled Shorbaji, Brett Welch, Arman Kilic.

**Resources:** Brett Welch, Arman Kilic.

**Software:** Brett Welch.

**Supervision:** Arman Kilic.

**Validation:** Curry Sherard.

**Writing – original draft:** John Del Gaizo, Curry Sherard.

**Writing – review & editing:** John Del Gaizo, Curry Sherard, Arman Kilic.

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
