## [Decision Letter · Decision Letter 0]

21 Sep 2023

PONE-D-23-27712Artificial Intelligence Predicts Coronary Artery Bypass Graft Outcomes Using a Single Surgical NotePLOS ONE

Dear Dr. Kilic,

Thank you for submitting your manuscript to PLOS ONE. After careful consideration, we feel that it has merit but does not fully meet PLOS ONE’s publication criteria as it currently stands. Therefore, we invite you to submit a revised version of the manuscript that addresses the points raised during the review process.

We look forward to receiving your revised manuscript.

Kind regards,

Amirmohammad Khalaji

Academic Editor

PLOS ONE

Journal Requirements:

Reviewers' comments:

Reviewer's Responses to Questions

**Comments to the Author**

1. Is the manuscript technically sound, and do the data support the conclusions?

Reviewer #1: Yes

Reviewer #2: Yes

Reviewer #3: Yes

2. Has the statistical analysis been performed appropriately and rigorously? 

Reviewer #1: Yes

Reviewer #2: I Don't Know

Reviewer #3: I Don't Know

3. Have the authors made all data underlying the findings in their manuscript fully available?

Reviewer #1: No

Reviewer #2: Yes

Reviewer #3: Yes

4. Is the manuscript presented in an intelligible fashion and written in standard English?

Reviewer #1: Yes

Reviewer #2: Yes

Reviewer #3: Yes

5. Review Comments to the Author

Reviewer #1: I would like to thank the authors for studying the outcomes following CABG using AI approaches. It is an interesting paper. However, there are some comments as follows:

Title: It is recommended to change the title to “Prediction of Coronary Artery Bypass Graft Outcomes Using a Single Surgical Note: An Artificial intelligence-based prediction model study”

Introduction:

Line 82-83: It is recommended to explain the superiority of AI to conventional analysis (e.g., logistic regression). To the best of my knowledge, AI models did not perform better. Please explain the statistical rational for this approach.

Line 84: In prediction model studies, it is better use predictors instead of risk factors. Please consider it in the whole manuscript, as well.

Methods: This section has been written well. But, according to the recent recommendation, it is recommended to write this section based on TRIPOD writing standard to improve the study replicability.

Please identify that isolated on-pump or off-pump CABG?

Line 195, SHAP value? Please provide its expanded form.

Line 119-120 Performance was measure by AUC. AUC provides information on discriminative ability of the models. It is recommended to provide the Calibration of the models, as an Achill heel in prediction model studies.

Please provide IRB or REC approval.

Results: line 212: median of AUC. It is better to use average instead.

Table 1: It is recommended to write Table 1 in a way to be understand easily by the readers. Please provide the expanded form of abbreviations bottom of the table.

Discussion:

Please discuss the results with previous studies in terms of AUC and predictors.

You can use the following large scale studies:

- Nomali, M., Heidari, M.E., Ayati, A. et al. Risk factors of in-hospital mortality for isolated on-pump coronary artery bypass graft surgery in the northeast of Iran from 2007 to 2016. Ir J Med Sci (2023). https://doi.org/10.1007/s11845-023-03298-6

- Sattartabar B et al (2021) Sex and age difference in risk factor distribution, trend, and long-term outcome of patients undergoing isolated coronary artery bypass graft surgery. BMC Cardiovasc Disord 21(1):1–10

- Also, please provide study limitations and implication.

Reviewer #2: Sherard et al. have conducted a study on designing AI models for the prediction of CABG outcomes in a single center. The findings are interesting; however, there are concerns that should be addressed.

- Features used for training the models should be more explained since they are not clear to the readers. What do the authors mean by surgical notes in the methods and also feature importance in the results section?

- The abbreviations should be mentioned in complete form in their first use (e.g., IQR).

- The headings “ Permutation Tests”, “Embedding Correlations”, and “Feature Importance” in the results section should be further elaborated.

- In the discussion section, a comparison should be made with other AI studies performed on patients undergoing CABG (such as 10.3389/fcvm.2022.977747, 10.3389/fcvm.2022.881881, and 10.1016/j.athoracsur.2021.08.040)

- The study lacks a limitation section which should be mentioned before the conclusion.

Reviewer #3: Summary

The authors have presented two Convolutional Neural Network (CNN)-based models designed to predict Coronary Artery Bypass Grafting (CABG) outcomes utilizing a single pre-operative surgical note. The first model, TextCNN, is a previously known CNN architecture specifically tailored for processing textual input, while the second model (AttnToNumCNN) represents an enhanced CNN architecture that incorporates a novel embedding pipeline for encoding numerical values. Furthermore, the enhanced model incorporates a multi-headed self-attention module. Notably, the authors have demonstrated good predictive performance with Area Under the Receiver Operating Characteristic Curve (AUC) values of 0.79 and 0.82 for TextCNN and the enhanced model, respectively.

Strengths:

* New numerical data embedding pipeline: The introduction of a new embedding layer for encoding numerical data within the medical context is a notable contribution.

* Performance on the primary task: The study has successfully achieved strong predictive performance on the primary task, as evidenced by the AUC values, underscoring the effectiveness of the proposed models in predicting CABG outcomes.

* SHAP Value Utilization: The incorporation of SHAP (Shapley additive explanations) values to explain the model's predictions is a commendable approach, fostering interpretability and transparency in the model's decision-making process.

Weaknesses:

* Lack of Clarity Regarding New Modules: The manuscript could benefit from a more explicit presentation of the specific contributions and impacts of the novel modules introduced in the enhanced model.

* Limited Model Exploration: The authors have primarily explored CNN-based architectures in this study. However, considering the sequential nature of medical notes, it would be advantageous to experiment with alternative models, such as Long Short-Term Memory (LSTM) or Transformer-based architectures, which are designed for effective long-sequence processing.

Issues

Major:

1. The authors have incorporated two novel modules, namely the embedding layer and the self-attention module, into their AttnToNumCNN architecture. They subsequently compare its performance to the baseline TextCNN model. However, a key issue arises in disentangling the distinct contributions of each of these modules to the observed performance enhancement. This lack of clarity raises the possibility that the improved performance may be primarily attributed to the introduction of the self-attention module alone, even if used along with the original embedding layers of the TextCNN model. Furthermore, the introduction of both the embedding layer and self-attention module introduces additional trainable parameters into the enhanced architecture, thereby complicating the fair comparison with the TextCNN model. To address this concern, it is advisable for the authors to report the total number of parameters for both models (it would be great if the number of parameters of the new modules are reported separately as well) and perform an ablation study by introducing only the new embedding layer, subsequently reporting its net impact on performance and then doing the same after adding the self-attention layer. This approach would help disentangle the individual contributions of these modules and enhance the interpretability of the performance improvements observed.

Minor:

1. Utilize Alternative Text Models: While the authors justified their omission of larger models like BERT due to computational costs and literature suggesting comparable performance with smaller models, it is advisable to explore traditional text-processing model families such as RNN and LSTM. These architectures are better suited for processing long-sequences compared to CNNs, which mainly focus on local patterns. Additionally, considering the dominance of Transformer-based architectures in NLP tasks, incorporating a scaled-down Transformer variant with a comparable parameter count to TextCNN could provide valuable insights into their relative effectiveness.

2. Visualize Self-Attention Weights: It is recommended to visualize the self-attention weights for a few input examples to gain visual insights into the behavior of the attention layer. This visualization would enhance the comprehensibility of the attention mechanism's functioning, thereby contributing to a more thorough investigation of its role in model decisions.

6. PLOS authors have the option to publish the peer review history of their article (what does this mean?). If published, this will include your full peer review and any attached files.

Reviewer #1: **Yes: **Mahin Nomali, Ph.D in Epidemiology

Reviewer #2: No

Reviewer #3: No

---

## [Author Response · Author response to Decision Letter 0]

7 Feb 2024

The response.docx file addresses each of these excellent reviewer suggestions and concerns. The paper is significantly more thorough with a stronger argument now.

For data requests, the contact information for MUSC IRB is as follows:

URL: https://research.musc.edu/resources/ori/irb-contacts

Phone: 843-792-4148

Stacey Goretzka, CIP

Program Manager

843-792-6527

goretzka@musc.edu

Summer Young, JD, MPH, CIP

IRB Reliance Manager

843-792-4144

youngsn@musc.edu

---

## [Decision Letter · Decision Letter 1]

26 Feb 2024

PONE-D-23-27712R1Prediction of Coronary Artery Bypass Graft Outcomes Using a Single Surgical Note: An Artificial Intelligence-based Prediction Model StudyPLOS ONE

Dear Dr. Kilic,

Thank you for submitting your manuscript to PLOS ONE. After careful consideration, we feel that it has merit but does not fully meet PLOS ONE’s publication criteria as it currently stands. Therefore, we invite you to submit a revised version of the manuscript that addresses the points raised during the review process. Minor revisions are required according to reviewer 1's comments.

We look forward to receiving your revised manuscript.

Kind regards,

Amirmohammad Khalaji

Academic Editor

PLOS ONE

Journal Requirements:

Reviewers' comments:

Reviewer's Responses to Questions

**Comments to the Author**

1. If the authors have adequately addressed your comments raised in a previous round of review and you feel that this manuscript is now acceptable for publication, you may indicate that here to bypass the “Comments to the Author” section, enter your conflict of interest statement in the “Confidential to Editor” section, and submit your "Accept" recommendation.

Reviewer #1: (No Response)

Reviewer #2: All comments have been addressed

Reviewer #3: All comments have been addressed

2. Is the manuscript technically sound, and do the data support the conclusions?

Reviewer #1: Yes

Reviewer #2: Yes

Reviewer #3: (No Response)

3. Has the statistical analysis been performed appropriately and rigorously? 

Reviewer #1: Yes

Reviewer #2: Yes

Reviewer #3: (No Response)

4. Have the authors made all data underlying the findings in their manuscript fully available?

Reviewer #1: Yes

Reviewer #2: (No Response)

Reviewer #3: (No Response)

5. Is the manuscript presented in an intelligible fashion and written in standard English?

Reviewer #1: Yes

Reviewer #2: (No Response)

Reviewer #3: (No Response)

6. Review Comments to the Author

Reviewer #1: I would like to thank the authors for dedicating time to revise the paper accordingly. However, there are comments as follows:

Abstract, methods: Please provide the expanded form of MM at first.

Abstract, results: I could not understand why you provided IQR for ROC instead of 95% CI. Please provide AUC and its uncertainty (95%CI).

Introduction:

Please provide the expanded form of AI, firstly. Please consider it in the whole document.

How about Research ethics approval? And, calibration as a performance measure for prediction model studies?

Results:

You can not report mean and IQR. You can report Mean and SD or Median and IQR. In addition, you must report AUC and 95 % Confidence interval, instead.

This section has been written in a complex way that may be it is interested for statistician or epidemiologist NOT clinicians. It recommended to use graph or table to facilitate its understanding.

Reviewer #2: (No Response)

Reviewer #3: (No Response)

7. PLOS authors have the option to publish the peer review history of their article (what does this mean?). If published, this will include your full peer review and any attached files.

Reviewer #1: **Yes: **Mahin Nomali

Reviewer #2: No

Reviewer #3: No

---

## [Author Response · Author response to Decision Letter 1]

4 Mar 2024

Reviewer 1

I would like to thank the authors for dedicating time to revise the paper accordingly. However, there are comments as follows:

Abstract, methods: Please provide the expanded form of MM at first.

* The abstract methods section has been changed accordingly.

Abstract, results: I could not understand why you provided IQR for ROC instead of 95% CI. Please provide AUC and its uncertainty (95%CI).

* This is a good point. We now report the mean AUC and the 95% CI.

Introduction:

Please provide the expanded form of AI, firstly. Please consider it in the whole document. 

* The first mentions of AI in the Introduction, Results, and Abstract sections are now expanded first.

How about Research ethics approval? And, calibration as a performance measure for prediction model studies?

* We clarified that “the study was deemed exempt from review by the institutional review Board (IRB Pro00122587), the committee who oversees research ethics at the Medical University of South Carolina” in the end of the Data sub-section of “Materials and Methods”.

* To address calibration as a performance measure, we have added the sentence: “Our results indicate that low parameter-count neural networks can achieve superior predictive performance to the current risk calculator standard, given utilization of the novel numerical embeddings and calibration of model output to a desired class-separation threshold.”

Results:

You can not report mean and IQR. You can report Mean and SD or Median and IQR. In addition, you must report AUC and 95 % Confidence interval, instead.

* We now report mean, SD, and 95% confidence intervals.

This section has been written in a complex way that may be it is interested for statistician or epidemiologist NOT clinicians. It recommended to use graph or table to facilitate its understanding.

* Thank you for this comment. We have added three additional tables to improve feasibility of result visualization to this section.

Thank you for your time and these helpful revisions. They greatly improved the quality of the paper in our opinion. We hope these updates addressed the reviewer concerns.

John Del Gaizo and Curry Sherard

---

## [Editor Report · Decision Letter 2]

6 Mar 2024

Prediction of Coronary Artery Bypass Graft Outcomes Using a Single Surgical Note: An Artificial Intelligence-based Prediction Model Study

PONE-D-23-27712R2

Dear Dr. Kilic,

We’re pleased to inform you that your manuscript has been judged scientifically suitable for publication and will be formally accepted for publication once it meets all outstanding technical requirements.

Kind regards,

Amirmohammad Khalaji

Academic Editor

PLOS ONE
---

## [Editor Report · Acceptance letter]

11 Mar 2024

PONE-D-23-27712R2 

PLOS ONE

Dear Dr. Kilic, 

I'm pleased to inform you that your manuscript has been deemed suitable for publication in PLOS ONE. Congratulations! Your manuscript is now being handed over to our production team.

Kind regards, 

on behalf of

Dr. Amirmohammad Khalaji 

Academic Editor

PLOS ONE